# Dalbavancin in Bone and Joint Infections: A Systematic Review

**DOI:** 10.3390/ph16071005

**Published:** 2023-07-15

**Authors:** Sofia Lovatti, Giorgio Tiecco, Alice Mulé, Luca Rossi, Anita Sforza, Martina Salvi, Liana Signorini, Francesco Castelli, Eugenia Quiros-Roldan

**Affiliations:** 1Unit of Infectious and Tropical Diseases, Department of Clinical and Experimental Sciences, University of Brescia and ASST Spedali Civili di Brescia, 25123 Brescia, Italy; s.lovatti001@unibs.it (S.L.); g.tiecco@unibs.it (G.T.); alice.mule@outlook.it (A.M.); l.rossi029@unibs.it (L.R.); a.sforza@unibs.it (A.S.); m.salvi026@unibs.it (M.S.); francesco.castelli@unibs.it (F.C.); 2Unit of Infectious and Tropical Diseases, ASST Spedali Civili di Brescia, 25123 Brescia, Italy; liana.signorini@unibs.it

**Keywords:** dalbavancin, DBV, osteomyelitis, spondylodiscitis, arthritis, osteoarticular infections, bone, joints, Gram-positive, Systematic review, review

## Abstract

Background: Approved for acute bacterial skin and skin structure infections, dalbavancin (DBV) has gradually acquired over the years a role as an off-label treatment for several infections caused by Gram-positive bacteria even in other anatomical sites. Osteoarticular (OA) infections are one of the most difficult-to-treat infections and, since the absence of recommendations, clinicians use different and heterogenic DBV dosing schedule regimens for the off-label treatment of osteomyelitis, spondylodiscitis, and septic arthritis. Our aim is to systematically review the current literature to describe DBV administration schedules and their outcome in OA infections. Methods: According to the 2020 updated PRISMA guidelines, all peer-reviewed articles regarding the use of DBV in OA infections were included. We conducted a literature search on PubMed and Cochrane Controlled Trials. Results: A total of 23 studies and 450 patients were included, prevalently male (144/195, 73.8%) and diabetic (53/163, 32.5%). Overall, 280 (280/388, 72.2%) osteomyelitis, 79 (79/388, 20.4%) spondylodiscitis, and 29 (29/388, 7.5%) septic arthritis were considered. *Staphylococcus aureus* (164/243, 67.5%) was the most common pathogen isolated. A previous treatment failure (45/96, 46.9%) was the main reason for a switch to a long-acting antibiotic. Most patients were successfully cured with DBV (318/401, 79.3%). A source control was performed in most patients with a favourable outcome (80.4%), while MRSA was prevalently isolated in people with an unfavourable outcome (57%). While a higher percentage of success was found in people who received three doses of DBV 1 week apart (92.3%), a higher rate of treatment failure was recorded in cases of when the DBV cycle was composed of less than two or more than four doses (27.8%). Conclusions: DBV has shown to be effective as a treatment for OA infections. The most favourable outcome was found in patients receiving three doses of DBV and with an adequate surgical management prior to antibiotic treatment. Although a rigorous administration schedule does not exist, DBV is a viable treatment option in the management of OA infections.

## 1. Introduction

Dalbavancin (DBV) is a long-acting parenteral lipoglycopeptide LGP. It was first approved by the Food and Drug Administration (FDA) in 2014 for acute bacterial skin and skin structure infections (ABSSSI), DBV has gradually acquired over the years a role as an off-label treatment for several infections caused by Gram-positive bacteria [1,2]. The mechanism of action of DBV involves the inhibition of peptidoglycan cross-linking in the bacterial cell wall [3]. DBV carries out potent activity against a broad spectrum of Gram-positive pathogens, including *Staphylococcus aureus* (SA), coagulase-negative staphylococci (CoNS), enterococci, and streptococci [4]. Regarding multi-drug resistant (MDR) pathogens, DBV is active against methicillin-resistant *S. aureus* (MRSA) and non-*VanA* MDR enterococci [4,5,6].

DBV has a long half-life of around 14 days and high plasma protein binding (approximately 93%) [7]. The is no need to adjust the dose in cases of hepatic impairment or mild to moderate renal impairment, and the absence of an interaction with cytochrome enzyme activity contributes to its easy handling [8]. Moreover, age has no impact on the pharmacokinetics profile of DBV [8]. The pharmacokinetics (PK) and pharmacodynamics (PD) parameters that best correlate with the in vivo efficacy of DBV, is the ratio of the mean free area under the curve to the minimum inhibitory concentration ratio (fAUC/MIC) [8,9]. Due to its PK properties, the use of DBV allows an early discharge and an easier outpatient management even without the necessity of maintaining intravenous access [10]. Additionally, due to its effectiveness in infections caused by MDR bacteria, which are increasing over time, the use of DBV for the treatment of several infections, such as bloodstream infections (BSI), endocarditis, prosthetic-joint infections (PJIs), arthritis, and osteomyelitis, is widely spreading [8].

Osteoarticular (OA) infections are one of the most difficult-to-treat infections due to the reduced penetration of many antibiotics into bone and joint structures, and the challenge of eradicating bacteria embedded in the biofilm state [11,12,13]. Both factors determine the need for long-course therapies and, frequently, for surgical interventions [12,13,14]. *Staphylococcus aureus* is the pathogen most frequently involved in bone and joint infections, followed by CoNS, enterococci, and streptococci [14]. Due to the continuous increase in MDR germs, the treatment of OA infections is becoming even more challenging [1]. Efficacy and effectiveness of DBV in OA infections are amply demonstrated in the available literature, which includes an in vitro study [3], one randomized clinical trial [15], and several real-life experiences [11,16,17,18].

However, despite its proven effectiveness for OA infection treatment, DBV is approved only for ABSSSIs. The current recommended regimen for the treatment of skin and soft tissue infections consists of a dose of 1500 mg, which may be administered as a single infusion of 1500 mg or fractionated into two doses of 1000 mg and 500 mg, respectively, 1 week apart [1]. In the absence of recommendations, different and heterogenic DBV dosing schedule regimens are used by clinicians for the off-label treatment of osteomyelitis, spondylodiscitis, and septic arthritis [11]. To the best of our knowledge, no meta-analysis or systematic reviews are to date available regarding the best administration schedule of DBV in cases of osteoarticular (OA) infections. Our aim is to systematically review the current literature to describe DBV administration schedules and their outcomes in OA infections.

## 2. Materials and Methods

Our methods meet the Preferred Reporting Items for Systematic Reviews and Meta-Analysis (PRISMA) updated guideline for systematic review stated in 2020 [19].

### 2.1. Eligibility Criteria

All randomized clinical trials (RCTs), prospective studies, retrospective studies, case series, or case reports published in peer-reviewed medical journals, regarding the use of DBV in bone and joint infections were included. We excluded articles regarding other antibiotics, non-osteoarticular-related infections (including prosthetic joint infections and hardware-related osteomyelitis), or papers in which data, regarding the use of DBV in bone and joint infections, were included in a wider database, and, therefore, impossible to extrapolate. Also, we excluded articles published in non-English languages, pre-print or ahead of print analysis, pre-clinical studies (including in vitro or animal model studies), reviews, systematic reviews, metanalysis, short communications, letter to the editor, and commentaries.

### 2.2. Information Sources and Search Strategy

An electronic search was employed to find the published articles which reported OA infections treated with DBV through the United States National Library of Medicine, PubMed (last accessed February 2023), MED-LINE (last accessed February 2023), PubMed Central, PMC (last accessed February 2023), and the Cochrane Controlled Trials (last accessed February 2023). References for this review were identified with the following research terms combination: “Dalbavancin” AND “Bone” OR “Bones” OR “Osteoarticular” OR “Osteomyelitis” OR “Spondylitis” OR “Spondylodiscitis” OR “Arthritis” OR “Joint” OR “Joints”. No time window was applied to the search.

### 2.3. Selection and Data Collection Process

A team of six resident doctors in Infectious and Tropical Diseases of the University of Brescia, Italy, read the abstract of each scientific work and independently selected the articles according to the established criteria (GT, AM, MS, SL, AS, LR). A Professor in Infectious and Tropical Diseases of the University of Brescia, Italy (EQR), revised the included and the rejected papers. Then, the selected papers were equally distributed among each resident doctor in order to assess for eligibility by reading the full text manuscript. Each resident doctor read, collected, and synthesised the data for the articles assigned using a detailed database. Afterwards, a random reassignment of the articles was carried out and each doctor reviewed the data collected by colleagues. Disagreements were resolved by a joint discussion supervised by the Professor in Infectious and Tropical Diseases.

### 2.4. Data Items

For each selected article, we collected information regarding the number of patients treated with DBV for OA infections, their demographic (age, sex, and ethnicity) and anamnestic (comorbidities, renal function, first episode/relapse, and known colonization) data. Focusing on infections, we reported the involved bones or joint, and the diagnostic tools performed to assess the diagnosis (biopsy/instrumental). The aetiological data, comprehending the type of sample, the results of the cultures and the resistance profiles, if any, were collected. The need of previous surgical intervention or a previous course of antibiotics and the reasons for any antibiotics-switches, were reported. We collected the scheme of DBV used (specifying posology of dose, number of doses, and length of the time intervals), and the eventual concomitant use of other antibiotic treatments. Furthermore, we reported on the outcome at the first evaluation after treatment and at the end of the follow-up period. Eventual drug adverse events and changes in patients’ quality of life (QoL) were collected. Missing or unclear data were reported as “non-available”. During the process of collecting data, we merged the pathogens which were outside the spectrum of DBV into a comprehensive category named “Other”. At the same time, we uncollected data regarding the use of uncategorizable drugs, such as caspofungin or metronidazole (used each just in one case) and, therefore, deemed irrelevant for the analysis.

## 3. Results

### 3.1. Study Selection and Search Results

A total of 109 papers were identified through our search. We excluded two duplicate and two non-English articles. A further 61 analyses were removed as 42 were systematic reviews or non-clinical articles and 19 were pre-clinical studies. Lastly, seven papers were excluded as their abstract did not meet the inclusion criteria. The remaining 37 articles were assessed for eligibility by a full-text analysis. Eleven were excluded as data, regarding bone and joints infections, were impossible to select and three further analyses were removed as no clinical use of DBV was described. Eventually, 23 studies were included as shown in the following flow diagram (Figure 1).

Most studies were retrospective non-randomized (15/23, 65.2%) and case reports (4/23, 17.4%). Only one RCT was present (1/23, 4.3%). Regarding the geographic distribution of the studies, 60.9% of the articles were from Europe (14/23). The study characteristics, number of patients, type of infection, aetiologic data, outcome, and the DBV schedule of administration are summarized in Table 1.

### 3.2. Results of Synthesis

A total of 450 patients were included. Considering the available demographic data, patients were mostly male (144/195, 73.8%), and with a mean age of 57.2 years old (standard deviation ± 8.4). The most common comorbidities identified were diabetes mellitus (53/163, 32.5%), hypertension and other cardiovascular diseases (40/163, 24.5%), renal diseases (11/163, 6.7%), and malignancies (8/163, 4.9%). Six (6/163, 3.7%) patients were otherwise healthy. Overall, 280 (280/388, 72.2%) osteomyelitis and 79 (79/388, 20.4%) spondylodiscitis were included, mainly involving the lower extremities (123/148, 83.1%) and lumbosacral vertebrae (21/30, 70%). A further 29 (29/388, 7.5%) septic arthritis were included, as shown in Table 2. Of the available data, most of the osteoarticular infections represented a first episode (46/48, 95.8%) rather than a recurrence (2/48, 4.2%). A total of 243 aetiologic agents were considered. *Staphylococcus aureus* (164/243, 67.5%) was the most common pathogen isolated, followed by CoNS (28/243, 11.5%), *Enterococcus* spp. (12/243, 4.9%), and *Streptococcus* spp. (9/243, 3.7%). These pathogens were isolated mostly from bone or deep tissues biopsies (110/130, 84.6%).

As regards the treatment before switching to a long-acting antibiotic, a surgical source control of the infection site was frequently performed before the use of DBV (110/159, 69.2%). In about one third of the cases no other antibiotics were administered before DBV (83/239, 34.7%), while, in case of previous lines of antimicrobial treatment, an anti-Gram-positive molecule was generally mostly used (62/239, 25.9%). As shown in Table 3, a previous treatment failure (45/96, 46.9%) was the main reason for switching to a long-acting antibiotic. DBV was mostly well tolerated with no adverse reactions (ADR) reported (255/285, 89.5%). Among the ADR, the most frequent were cutaneous allergic reactions, followed by nausea and diarrhoea. Additionally, considering the available data, no cases of *Clostridioides difficile* infections were reported. As shown in Table 3, other concomitant antibiotics were rarely used (56/238, 23.5%) during DBV treatment. While a prevalently anti-Gram-positive spectrum was covered by pre-DBV antimicrobials, most of the co-administered regimen were both anti-Gram-positive and anti-Gram-negative (23/238, 9.7%) as shown in Figure 2.

The scheme of administration and doses of DBV were fully described in 320 patients (71.1%) as shown Table 4. Two doses of DBV administered exactly 1 week apart (184/320, 57.5%) were the most common therapeutic scheme used. However, four doses or more cycles (76/320, 23.8%) were also recorded.

Most patients were considered successfully cured after DBV (318/401, 79.3%) as most of the patients did not need any other antibiotic treatment after DBV (180/204, 88.2%). Treatment failure was recorded in 18.7% of cases (75/401, 18.7%) for persistent infection (44/401, 11.0%), relapse (21/401, 5.2%), or the need to switch therapy (10/401, 2.5%). Between the pharmacological failures, we underline that 70.8% of patients who needed surgery after treatment with DBV (24/231, 10.4%) underwent a non-conservative intervention, such as amputation. A consecutive antibiotic treatment was prescribed in a few patients, (17/204, 8.3%), in seven of them (7/17, 41.2%) the chosen molecule had the same spectrum of activity as DBV. Of the available data, follow-up visits within and after a 4 week free-treatment period enlightened a high percentage of clinical success, respectively, of 84.3% (231/274) and 77.4% (212/274) as shown in Table 5.

Table 6 compares different aspects in patients with favourable or unfavourable outcome. An outcome was defined as favourable in patients considered successfully cured at their last visit, while unfavourable outcomes included persistent infections, relapses, therapeutic switches, and infection-related deaths. A source control was performed in most patients with favourable outcomes (80.4%), while MRSA was prevalently isolated in people with unfavourable outcomes (80%). As shown in Figure 3, while a higher percentage of success was found in people who received three doses of DBV 1 week apart (92.3%), a higher rate of treatment failure was recorded in cases when the DBV cycle was composed of less than two doses. However, the highest rate of DBV failure was described in patients who received more than four doses (27.8%), perhaps reflecting a prolonged rescue schedule in critical and difficult-to-treat patients.

## 4. Discussion

This systematic review tries to describe the DBV administration schedules and their outcome in OA infections. Bone and joint infections are among the most difficult-to-treat infections, representing a clinical challenge not only for the infectious disease specialist. In this context, the use of DBV is quickly spreading, although there is an absence of specific dosing recommendations and on-label prescriptiveness [38]. To date, OA infections are the most frequent off-label indication for DBV [8,39]. This long-acting antibiotic not only is effective in the management of Gram-positive infections, requiring prolonged therapy, but it also contributes to shorten the hospital length-of-stay, decrease treatment-related costs, and improve patients’ quality of life [30,40,41,42,43,44]. Therefore, the use of DBV may have major significance within the current antimicrobial stewardship programmes to reduce the length of ospitalization.

Our analysis shows that the main reason for switching to DBV was the failure of a previous antimicrobial treatment regimen. Clinicians should always be aware that, in case of pharmacological failure, an appropriate source control should be considered: the correlation between appropriate bone debridement and a better outcome is widely demonstrated in the literature [12,14,45]. In our study, 67.5% of the microorganisms isolated were *S. aureus* and, considering only those with a known resistance profile, 47.4% of them were MRSA. The prescription of a long-acting molecule allows the clinician to continue long-term treatment as an outpatient, reducing costs and hospitalization-related complications [40,46]. As it is shown by recently published articles, DBV should be considered, in view of an antimicrobial simplification [22], to reduce medication burden and to encourage an early discharge strategy [46].

Our results also underline that most of the patients (65.3%) received antibiotics before DBV, mostly with an anti-Gram-positive coverage. On the other side, only a quarter (24%) of patients received co-administered antimicrobial drugs together with DBV: this suggests that DBV is generally considered a reliable monotherapy choice for OA infections [47]. In this view, the rationale of adding a drug to DBV to broaden the spectrum of the long-acting therapy might be considered in case of infections with no possibility of device removal [47].

In our systematic review, 79.3% of patients were successfully treated with DBV, confirming the known efficacy, effectiveness, and safety of this antibiotic in OA infections [15,39,48]. Of the patients who failed, most showed a persistent of infection and the most significant difference between those with favourable and unfavourable outcomes was undergoing surgery before the antibiotic treatment. This shows that antimicrobial therapy alone is often not effective in treating infected and necrotic bone [12,14]. Removal of the necrotic material with formed biofilm and fluid collections, sometimes with eventual revascularization, is confirmed to be a crucial step in the management of osteomyelitis because they allow for a better antibiotic penetration into tissues [12,14,49,50]. Whilst MRSA was more represented in patients with unfavourable outcomes, although real world experience with combination therapy is very limited, the synergistic effect of DBV in combination with other antimicrobials, such as linezolid, ceftaroline, and rifampin, in reducing MIC for MRSA and heterogeneous vancomycin-intermediate (hVISA) strains has been proved in different in vitro studies [8,51,52,53]. Further studies are needed to assess the relevance of combination therapies in difficult-to-treat OA infections.

The aim of this systematic review was to describe the administration scheduled regimen and outcomes of DBV in the treatment of OA infections. The rational of Dalbavancin use in OA infections is based on its bone and joint penetration and its PK/PD properties, particularly in relation with renal clearance. Regarding the first topic, Dunne et al. [3] in a preclinical phase I study, demonstrated the good penetration of DBV in cortical bone and joint tissues. After a single dose of 1000 mg, the DBV concentrations in cortical bone and synovial fluid 12 h after infusion and 2 weeks later were 6.3 microg/g and 4.1 microg/g for bone and 22.9 microg/mL and 6.2 microg/mL in synovial fluid, respectively [3]. In the same study, models of quantitative exposure to DBV were performed using the regimen of two 1500 mg intravenous infusions 1 week apart. According to the data collected, that regimen would result in an exposure to DBV equal to or greater than the DBV MIC 99.9 for *S. aureus* for 8 weeks [3,6]. Additionally, the mean cortical bone to plasma AUC penetration ratio of DBV calculated by the authors was 13.1%, which is like the estimated free-in serum-DBV, the one with antimicrobial activity [3]. Furthermore, DBV demonstrated potent activity against the biofilm produced by *Staphylococcus* spp., and *Enterococcus* spp., at concentrations similar to those achieved in osteoarticular tissues [5,54,55]. In those in vitro preclinical studies, DBV was overall superior to the comparator vancomycin [54,55]. Cojutti et al. conducted a pharmacokinetic analysis in a prospective cohort population with OA infections caused by Gram-positive bacteria and treated with DBV [11]. With the dosing regimen used (two 1500 mg doses 1 week apart on days 1 and 8), Monte Carlo simulations showed that, considering the fAUC24h/MIC as the best estimate of efficacy, this dosing regimen provides excellent results against both MRSA and MSSA in the population in this study for a period of 5 weeks. This duration can be further extended to 7–9 weeks by administering an additional dose of 500–1500 mg on day 36 [11]. Paying attention to the relation between renal clearance and DBV exposure, in another study, Cojutti et al., conducted a population pharmacokinetic analysis of DBV use for several *S. aureus* infections [27]. The Monte Carlo simulation, with two 1500 mg doses 1 week apart, showed that the target plasma concentration of DBV was achieved for a period of up to 4, 5, or 6 weeks depending on the creatinine clearance classes, and particularly for clearances between 90–120 mL/min, 60–90 mL/min, and 30–60 mL/min, respectively [27]. At the same time, two 1000 mg doses 1 week apart showed that the target was achieved for up to 5 weeks in cases of a creatinine clearance of less than 30 mL/min [27]. This analysis showed that a shorter exposure duration of DBV results in people with a normal renal function than in people with a moderate renal impairment and that an individual variability in DBV clearance might be seen even among subjects with the same clearance class [27]. The pharmacokinetic profile of dalbavancin in people with renal impairment was already described in 2009 by Marbury et al., showing that exposure to the drug in people with moderate and severe renal impairment increased by 50 and 100%, respectively [56]. In this regard, therapeutic drug monitoring (TDM) of DBV would be a crucial tool in identifying the most appropriate time after which to administer doses following the first two, but nowadays this test is unavailable in any laboratories. Unfortunately, TDM data were not available almost in every scientific article included in our systematic review.

Consistently, our results show that the number of favourable clinical outcomes increases with the number of doses, up to three, after which the benefit decreases. The outcome improves as the number of administrations increases within the first three because infections of bone are known to require a high amount of antibiotic for a longer period [57,58]. On the other hand, the worsening of the outcome beyond three administrations, could be explained by a sub-optimal clinical management, perhaps a non-accurate source control or an inadequate antibacterial regimen in terms of spectrum or in terms of dosage might explain the persistence of the infection. Particularly, a low dose of DBV or a too wide interval between doses might not ensure sufficient levels of antibiotic exposure to enable proper treatment.

The findings of this systematic review should be seen in the light of some limitations. First, this review is mainly based on retrospective studies: only one RCT and three prospective studies were included. Although we have followed all PRISMA statements and guidelines for this systematic review, the protocol was not included in the PROSPERO platform. We know that registration is not mandatory but it ensures an anticipated and clear research plan; therefore, it could be considered another limit in the present analysis. In addition, the inclusion of retrospective studies describing aggregate data makes it difficult to select and analyse data for each individual patient. Furthermore, the heterogeneity of the included studies, each different in its design, provided us with a large subset of data but sometimes incomplete in specific aspects. For this reason, some sub-analyses consider only smaller datasets. In an effort to focus on the DBV dosing schedule and outcomes, we recorded the available data even though data on the specific type of OA infection, such as arthritis or osteomyelitis, were missing. For this reason, the number of patients included is greater than the sum of the different types of infection. In addition, very few data were available on polymicrobial infections, and when present, they were aggregate data. Thus, another limitation of our work is that more microorganisms were isolated than patients, which resulted in the absence of information on polymicrobial infections. Lastly, the heterogeneity of the studies included, in the absence of methods to assess the risk of bias or certainty, in the body of evidence restricted our review to a descriptive analysis. However, to the best of our knowledge, and according to the sources analysed, this would be the largest available collection of data concerning the use of DBV in the treatment of osteoarticular infections.

## 5. Conclusions

DBV has shown to be effective as a treatment for osteoarticular infections. Adequate surgical management prior to antibiotic treatment proved to be a determining factor in a good outcome in our patient cohort. The most favourable outcome was found in patients receiving three doses of DBV; our systematic review shows that clinicians should be aware in cases of MRSA OA infections and inadequate source controls. The utilization of four or more doses of DBV might not be associated with an improved outcome. Implementation of a test for TDM of DBV in laboratories could allow the best customized timing to be identified for a third dose administration. In the absence of that, attention must be paid to the renal function class for choosing the best distance between the doses. Although the on-label prescription of this new molecule includes only ABSSSI, currently, many specialists choose DBV as a first-line treatment for Gram-positive infections of the osteoarticular. To date, a rigorous administration schedule does not exist; however, DBV is a viable treatment option in the management of OA infections. In light of these considerations, dalbavancin appears to be a promising option for treating OA infections from a clinical, microbiological, pharmacological, and pharmacoeconomic point of view, allowing for an early discharge of the patient owing to its long-term mode of action and contributing to the antimicrobial stewardship program.

## Figures and Tables

**Figure 1 pharmaceuticals-16-01005-f001:**
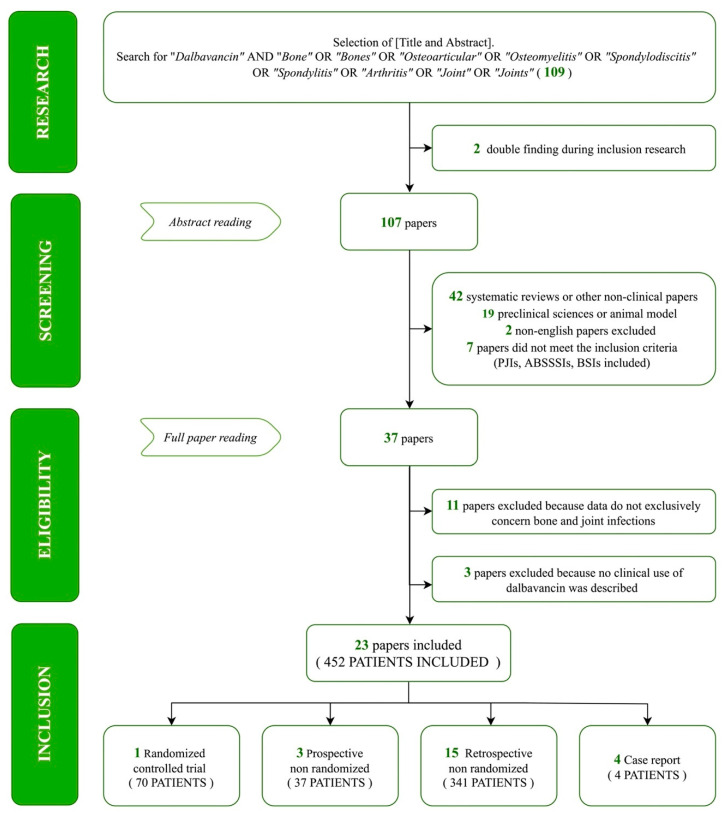
Research term combinations and flow-chart of the study selection process. Acronym used—PJIs: prosthetic joint infection, ABSSSIs: acute bacterial skin and soft tissues infections, BSIs: bloodstream infections.

**Figure 2 pharmaceuticals-16-01005-f002:**
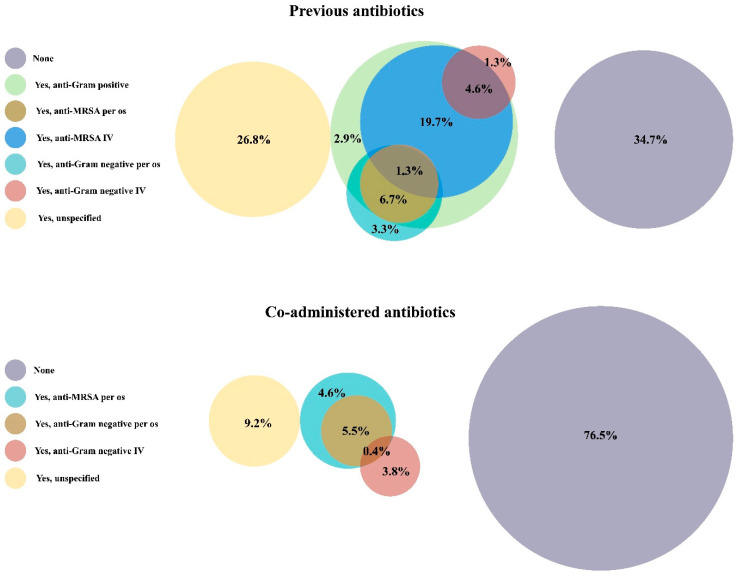
Venn diagrams comparing the prescription of antimicrobials before starting a DBV treatment (up) and while administering DBV (down). Every circle represents a combination of antibiotic molecules pooled based on their antibacterial spectrum of activity. The sizes of the circles are proportional to the number of patients that were prescribed with the corresponding therapeutic regimen. Combination therapies were considered for their global antibacterial spectrum. Percentages refer to the single category. Grouping was assessed as follows: anti-Gram-positive (cefazolin, oxacillin, amoxicillin/clavulanic acid); anti-MRSA po (rifampin, linezolid, tedizolid, clindamycin); anti-MRSA IV (vancomycin, daptomycin, teicoplanin, ceftaroline); anti-MRSA and anti-Gram-negative po (TMP/SMX, tetracycline, quinolone Anti-MRSA); and anti-Gram-negative IV (ceftriaxone, ertapenem, meropenem, gentamycin, fosfomycin, piperacillin/tazobactam).

**Figure 3 pharmaceuticals-16-01005-f003:**
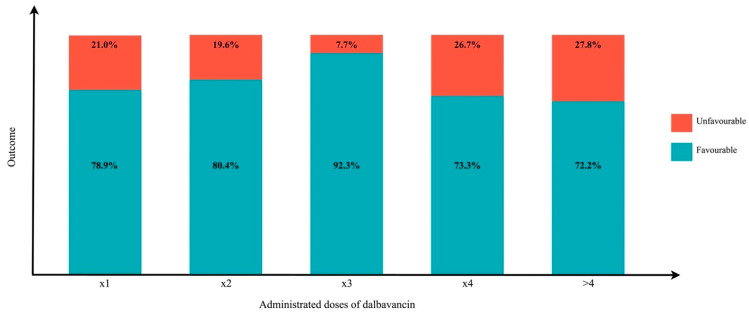
Favourable versus unfavourable outcomes in different administration schedules of DBV: in the histogram every column represents the total number of patients treated with a specific number of doses of DBV, regardless of the interval of administration. The percentages represented in red and blue refer, respectively, to unfavourable and favourable outcomes.

**Table 1 pharmaceuticals-16-01005-t001:** First author, number of patients, type of infection, aetiologic data, outcome, and dalbavancin schedule of administration of every included study analysed.

First Author	Ref.	N. of Patients	Type of the Study	Type of Infection (n)	Aetiology (n)	Outcome (n)	DBV Dosage (n)	N. Administrations (n)
Rappo U, et al.	[15]	70	Randomized controlled trial	Osteomyelitis (70)	*S. aureus* (42) CoNS (14) *Enterococcus* spp. (8) *Streptococcus* spp. (3) Other Gram-positive (13) Gram-negative and fungi (3)	Clinical cure (63) Clinical cure after different therapy (1) Lost to FU (2) Unrelated death (1) NA (3)	1500 mg (70)	1 dose (3) 2 doses, 1 week apart (67)
Morata L, et al.	[16]	19	Retrospective	Osteomyelitis (12) Arthritis or spondylodiscitis (7)	*S. aureus* (10) CoNS (4) *Enterococcus* spp. (2) *Streptococcus* spp. (3) Other Gram+ (2)	Clinical cure (17) Clinical failure (1) Relapse (1)	NA	NA
Almangour TA, et al.	[20]	27	Retrospective	Osteomyelitis (16) Spondylodiscitis (14)	*S. aureus* (27) CoNS (1) *Streptococcus spp.* (2) Gram-negative and fungi (1)	Clinical cure (24) Clinical failure (1) NA (2)	1500 mg (11) 1000 + 500 mg (15) 2000 + 500 mg (1)	1 dose (6) 2 doses, 1 week apart (6) 3 doses > 1 week apart (1) 3 doses, 1 week apart (2) 4 doses,1 week apart (4) 4 doses > 1 week apart (19) >4 doses (7)
Stroffolini G, et al.	[21]	25	Prospective	Osteomyelitis (9) Spondylodiscitis (6) Arthritis (3)	NA	Clinical cure (9) NA (16)	1500 mg (25)	1 dose (3) 2 doses, 1 week apart (9) NA (13)
Ramadan MS, et al.	[22]	13	Retrospective	Spondylodiscitis (13)	*S. aureus* (6) CoNS (4) Other Gram-positive (1) Gram-negative and fungi (2)	Clinical cure (8) Relapse (5)	1500 mg (12) 1000 + 500 mgfor 5 times (1)	2 doses, 1 week apart (12) >4 doses (1)
Brescini L, et al.	[23]	13	Retrospective	Osteomyelitis (8) Arthritis (5)	NA	NA (13)	NA	NA
De Nicolò A, et al.	[24]	10	Prospective	Osteomyelitis (4) Spondylodiscitis (3) Arthritis (3)	*S. aureus* (7) CoNS (2) *Streptococcus* spp. (1)	NA (10)	1500 mg (10)	1 dose (3) 2 doses, 1 week apart (7)
Navarro-Jiménez G, et al.	[25]	23	Retrospective	Osteomyelitis (23)	*S. aureus* (12) Other Gram-positive (7) CoNS (3) *Enterococcus* spp. (1)	Clinical cure (20) Clinical failure (3)	1000 + 500 mg for 5 times (8) 1000 + 500 + 500 + 500 mg (1) 1000 + 500 mg for 9 times (1) 1500 mg (9) 1000 mg (1) 750 mg (1) 1500 + 500 mg (2)	1 dose (5) 2 doses, 1 week apart (5) 3 doses, 1 week apart (2) 4 doses, 1 week apart (1) >4 doses (10)
Tobudic S, et al.	[26]	38	Retrospective	Osteomyelitis (20) Spondylodiscitis (14) Septic arthritis (4)	NA	Clinical cure (19) Clinical failure (7) Clinical cure after different therapy (8) NA (4)	1000 + 500 (6) 1500 + 1000 mg (28) 1500 + 1500 mg (4)	NA
Cain AR, et al.	[18]	42	Retrospective	Osteomyelitis (42)	*S. aureus* (23) NA (19)	Clinical cure (20) Relapse (13) Clinical failure (9)	1500 mg (42)	2 doses, 1 week apart (42)
Cojutti PG, et al.	[27]	27	Retrospective	Osteomyelitis (16) Spondylodiscitis (9) Arthritis (2)	NA	NA (27)	1000 mg (NA) 1500 mg (NA)	NA
Cojutti PG, et al.	[11]	2	Prospective	Spondylodiscitis (1) Arthritis (1)	*S. aureus* (1) NA (1)	Clinical cure (1) Relapse (1)	1500 mg (2)	2 doses, 1 week apart (2)
Lueking R, et al.	[28]	19	Retrospective	Osteomyelitis (15) Arthritis (4)	MSSA (1) NA (18)	Clinical cure (18) Relapse (1)	1500 + 1500 (1) NA (18)	2 doses, 1 week apart (1) NA (14)
Tuan JJ, et al.	[29]	23	Retrospective	Osteomyelitis (21) Arthritis (2)	NA	Clinical cure (21) Clinical failure (1) Death from BJIs (1)	NA	NA
Mazzitelli M, et al.	[30]	14	Retrospective	Spondylodiscitis (14)	*S. aureus* (14)	Clinical cure (14)	1500 mg (14)	3 doses >1 week apart (7) 4 doses > 1 week apart (5)
Almangour TA, et al.	[31]	9	Retrospective	Osteomyelitis (6) Spondylodiscitis (3)	*S. aureus* (9)	Clinical cure (9)	NA (9)	NA
Dinh A, et al.	[17]	48	Retrospective	NA	NA	Clinical cure (35) Clinical failure (11) NA (2)	1000 mg (2) 1500 mg (37) 1500 + 1000 mg (1) 1500 + 500 mg (2) 1000 + 500 + 500 + 500 mg (6)	1 dose (5) 2 doses > 1 week apart (4) 2 doses, 1 week apart (30) 3 doses, 1 week apart (1) 4 doses > 1 week apart (3) >4 doses (5)
Loupa CV, et al.	[32]	1	Case report	Osteomyelitis (1)	*Enterococcus* spp. (1)	Lost at FU (1)	1500 mg (1)	2 doses > 1 week apart (1)
Bork JT, et al.	[33]	14	Retrospective	Osteomyelitis (13) Arthritis (1)	NA	Clinical cure (6) Clinical failure (4) Lost to FU (3) Unrelated death (1)	NA	NA
Ritchie H, et al.	[34]	1	Case report	Spondylodiscitis (1)	*S. aureus* (1)	Clinical cure after different therapy (1)	1500 mg (1)	1 dose (1)
Almangour TA, et al.	[35]	1	Case report	Spondylodiscitis (1)	*S. aureus* (1)	Clinical cure (1)	1000 + 1000 + 500 mg for 6 times (1)	>4 doses (1)
Molina Collada J, et al.	[36]	1	Case report	Septic arthritis (1)	Other Gram-positive (1)	Clinical cure (1)	1500 mg (1)	1 dose (1)
Bryson-Cagn C, et al.	[37]	10	Retrospective	Osteomyelitis (4) Septic arthritis (3) Spondylodiscitis (3)	*S. aureus* (10)	Clinical cure (6) Clinical failure (3) NA (1)	1000 mg (5) 1500 + 1000 + 500 + 500 + 500 mg (1) 1000 + 500 mg (3) 1000 + 500 + 500 mg (1)	1 dose (5) 2 doses, 1 week apart (3) 3 doses, 1 week apart (1) >4 doses (1)

**Table 2 pharmaceuticals-16-01005-t002:** Localization of osteoarticular infections, sample types, and isolated pathogens. (*) MSSA 43.3%, MRSA 39%, and S. aureus with an unknown profile of resistance 17.7%; (**) CoNS oxacillin-resistant 14.2%, CoNS oxacillin-sensitive 7.1%, CoNS with unknown resistance profile 78.6%; (***) Vancomycin-Resistant Enterococcus (VRE) 0.4%. Acronym used—CoNS: Coagulase Negative Staphylococcus.

**Type of osteoarticular infections**	
N. of patients included	388 (86.2%)
Osteomyelitis (n, %)	280 (72.2%)
Spondylodiscitis (n, %)	79 (20.4%)
Septic arthritis (n, %)	29 (7.5%)
**Site of infection**	
N. of patients included	178 (39.6%)
Osteomyelitis (total)	148
Lower extremities (n, %)	123 (83.1%)
Upper extremities (n, %)	18 (12.1%)
Other (n, %)	7 (4.7%)
Spondylodiscitis	30
Lumbo-sacral (n, %)	21 (70%)
Cervical-thoracic (n, %)	8 (26.7%)
Both (n, %)	1 (3.3%)
**Isolated pathogens**	
N. of isolated included	243
*S. aureus* * (n, %)	164 (67.5%)
Other Gram-positive (n, %)	24 (9.9%)
CoNS ** (n, %)	28 (11.5%)
*Enterococcus* spp. *** (n, %)	12 (4.9%)
*Streptococcus* spp. (n, %)	9 (3.7%)
Gram-negative and fungi (n, %)	6 (2.5%)
**Sample type**	
N. of patients included	130 (28.9%)
Bone biopsy or deep tissues (n, %)	110 (84.6%)
Blood (n, %)	19 (14.6%)
Bone biopsy or deep tissues and blood (n, %)	1 (0.8%)

**Table 3 pharmaceuticals-16-01005-t003:** Medical and surgical treatment before or co-administered with DBV. (*) Surgical procedures included: toilette of infected wound, bone debridement, drainage of abscesses. Acronym used—DBV: dalbavancin; MRSA: methicillin-resistant Staphylococcus aureus; IV: intravenous; po: oral; INR: international normalized ratio.

**Surgery before DBV**	
N. of patients included	159 (35.3%)
Patients who underwent surgery before DBV* (n,%)	110 (69.2%)
Patients who did not undergo surgery before DBV (n,%)	49 (30.8%)
**Antibiotics pre-DBV**	
N. of patients included	239 (53.1%)
None (n,%)	83 (34.7%)
Unspecified molecule (n,%)	64 (26.8%)
Anti-MRSA IV (n,%)	47 (19.7%)
Anti-MRSA AND anti-Gram-negative po (n,%)	16 (6.7%)
Anti-MRSA AND anti-Gram-negative IV (n,%)	14 (5.9%)
Anti-MRSA po (n,%)	8 (3.4%)
Anti-Gram-positive (n,%)	7 (2.9%)
**Reason to switch to DBV**	
N. of patients included	96 (21.3%)
Failure of the previous antimicrobial regimen (n,%)	45 (46.9%)
Simplification (n,%)	19 (19.8%)
Adverse reaction (n,%)	14 (14.6%)
Other (n,%)	18 (18.8%)
**Adverse reactions to DBV**	
N. of patients included	285 (63.3%)
No adverse reactions (n,%)	255 (89.5%)
Adverse reaction (n,%):	30 (10.5%)
➢ Unspecified (n,%)	13 (4.6%)
➢ Cutaneous (n,%)	7 (2.5%)
➢ Nausea (n,%)	6 (2.1%)
➢ Diarrhoea (n,%)	2 (0.7%)
➢ INR abnormalities (n,%)	1 (0.4%)
➢ Acute kidney injuries (n,%)	1 (0.4%)
**Antibiotics co-administered**	
N. of patients included	238 (52.9%)
None (n,%)	182 (76.5%)
Unspecified molecule (n,%)	22 (9.2%)
Anti-MRSA AND anti-Gram-negative IV (n,%)	10 (4.2%)
Anti-MRSA AND anti-Gram-negative po (n,%)	13 (5.5%)
Anti-MRSA po (n,%)	11 (4.6%)

**Table 4 pharmaceuticals-16-01005-t004:** DBV scheme of administration: lines correspond to number of DBV doses and time between administrations, columns refer to dosages administered (expressed in mg). (*) 1 patient was treated with 2000 mg + 500 mg × 4; (**) 1 patient was treated with 1500 mg + 1000 mg + 500 mg × 3; (***) 1 patient was treated with 1000 mg × 2; (****) 1 patient was treated with 1000 mg + 1000 mg + 500 mg × 6.

DBV Scheme of Administration	*n*	1500 + 1500 * *n*	1500 + 1000 * *n*	1500 + 500 * *n*	1000 + 500 * *n*	750 + 750 * *n*	Total (320 Patients)
1 dose	*0*	26			6	0	32 (10.0%)
2 doses, More than 1 week apart	*1*	9	3	1	0	0	13 (4.1%)
2 doses, 1 week apart	*1*	177	0	2	5 ***	0	184 (57.5%)
3 doses, More than 1 week apart	*2*	7	0	0	1	0	8 (2.5%)
3 doses, 1 week apart	*2*	3	0	0	4	0	7 (2.2%)
4 doses, More than 1 week apart	*3*	10	17	0	3	0	30 (9.4%)
4 doses, 1 week apart	*3*	0	0	1 *	4	0	5 (1.6%)
> 4 doses	*x*	3	9	4 **	24 ****	1	41 (12.8%)

**Table 5 pharmaceuticals-16-01005-t005:** Outcome and Follow-up.

**Outcome at last visit**		
N. of patients included		401 (89.1%)
Success (n,%)		318 (79.3%)
Failure for persistent infection (n,%)		44 (11.0%)
Failure for relapse (n,%)		21 (5.2%)
Failure for need to switch therapy (n,%)		10 (2.5%)
Lost-to-follow-up (n,%)		5 (1.2%)
Infection-related death (n,%)		1 (0.2%)
Unrelated death (n,%)		2 (0.5%)
**Outcome within and after 4 weeks since the end of treatment**		
N. of patients included		12; 274 (60.9%)
	**Outcome within 4 weeks**	**Outcome after 4 weeks**
Success (n,%)	231 (84.3%)	212 (77.4%)
Failure-persistent infection (n,%)	37 (13.5%)	25 (9.1%)
Failure-relapse (n,%)	2 (0.7%)	20 (7.3%)
Failure-switch therapy (n,%)	0 (0.0%)	10 (3.6%)
Lost-to-follow-up (n,%)	4 (1.5%)	5 (1.8%)
Infection-related death (n,%)	0 (0.0%)	0 (0.0%)
Unrelated death (n,%)	0 (0.0%)	2 (0.7%)

**Table 6 pharmaceuticals-16-01005-t006:** Comparison of different features in patients with favourable or unfavourable outcomes. Acronym used—MRSA: methicillin-resistant Staphylococcus aureus; CoNS: Coagulase Negative Staphylococcus spp.

	**Favourable Outcome**	**Unfavourable Outcome**
N. of patients included	318	76
**Surgery before DBV**		
Patients with available data	112	16
Subjected to surgery before DBV (n,%)	90 (80.4%)	4 (25%)
Did not undergo surgery before DBV (n,%)	22 (19.6%)	12 (75%)
**Isolated pathogens**		
Patients with available data	68	10
MRSA (n,%)	39 (57.4%)	8 (80%)
MSSA (n,%)	14 (20.6%)	2 (20%)
Other Gram-positive (n,%)	6 (8.8%)	0 (0.0%)
CoNS (n,%)	6 (8.8%)	0 (0.0%)
*Enterococcus* spp. (n,%)	2 (2.9%)	0 (0.0%)
*Streptococcus* spp. (n,%)	1 (1.5%)	0 (0.0%)
**Type of osteoarticular infections**		
Patients with available data	242	59
Osteomyelitis	180 (74.4%)	41 (70.7%)
Spondylodiscitis	45 (18.6%)	14 (23.7%)
Septic arthritis	17 (7%)	4 (6.8%)

## Data Availability

Not applicable.

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
