# Peer review of "Dalbavancin in Bone and Joint Infections: A Systematic Review"

_pharmaceuticals, 2023, doi:10.3390/ph16071005_

Round 1

Reviewer 1 Report

The authors focus on the Dalbavancin in bone and joint infections: a systematic review. The manuscript is interesting. However, I have few queries before accepting this article.

1. Why authors focus on Dalbavancin antibiotics????

2. What about the mode of action of this drugs.

3. Why this antibiotics frequently used in joint infection????

4. I observed that around 10% adverse reaction... why???

5. Similarly relapse cases...discuss in details??

6. What about future prospects... is there any combination therapy. 

7. What is the reason this drugs is more effective against MRSA rather than MSSA... 

Reviewer 2 Report

The aim was to systematically review the current literature to describe the Dalbavancin (DBV) administration schedules and their outcome in Osteoarticular (OA) infections.  It should be underlined that It is the first systematic review focused on this issue.  The Authors showed that the number of favorable clinical outcomes increases with the number of doses up to 3, after which the benefit decreases.  They concluded that DBV is effective as a treatment for osteoarticular infections.

The Authors have presented sufficient data. The article is easy to read and logically structured.  

There are some  comments in the reviewer's opinion that should be taken under consideration by the Authors:

1.     Please add the Adverse Effects, which were connected with the treatment with DBV, maybe it will be good to add it to Table 1 as a separate column

2.     How often Clostridioides difficile-associated diarrhea (CDAD) was observed?

3.     Please add to table 1 -a column with the type of the study ( retrospective etc)

4.      What was the cost of treatment for patients with OA.

5.     Please add the ID number of registration in PROSPERO as it is a systematic review

Round 2

Reviewer 1 Report

Accepted

Reviewer 2 Report

The Authors included all my remarks and  in my opinion the paer may be published in the current form.